# Efficient Single Image Super-Resolution with Entropy Attention and Receptive Field Augmentation

## ABSTRACT

Transformer-based deep models for single image super-resolution (SISR) have greatly improved the performance of lightweight SISR tasks in recent years. However, they often suffer from heavy computational burden and slow inference due to the complex calculation of multi-head self-attention (MSA), seriously hindering their practical application and deployment. In this work, we present an efficient SR model to mitigate the dilemma between model efficiency and SR performance, which is dubbed **E**ntropy **A**ttention and **R**eceptive **F**ield **A**ugmentation network (**EARFA**), and composed of a novel entropy attention (EA) and a shifting large kernel attention (SLKA). From the perspective of information theory, EA increases the entropy of intermediate features conditioned on a Gaussian distribution, providing more informative input for subsequent reasoning. On the other hand, SLKA extends the receptive field of SR models with the assistance of channel shifting, which also favors to boost the diversity of hierarchical features. Since the implementation of EA and SLKA does not involve complex computations (such as extensive matrix multiplications), the proposed method can achieve faster nonlinear inference than Transformer-based SR models while maintaining better SR performance. Extensive experiments show that the proposed model can significantly reduce the delay of model inference while achieving the SR performance comparable with other advanced models.

## CCS CONCEPTS

• **Computing methodologies**; • **Reconstruction**;

## KEYWORDS

Deep learning, Super-resolution, Entropy attention, Shifting large kernel attention

## 1 INTRODUCTION

Efficient single image super-resolution (ESISR) stands as a crucial task in low-level computer vision community that aims at striking a good balance between SR performance and model efficiency, which is different from high-fidelity SISR methods [1–3]. Therefore, ESISR methods are typically more compatible with application scenarios with constrained resources, which is one of the reasons why it has become a research hotspot in the field.

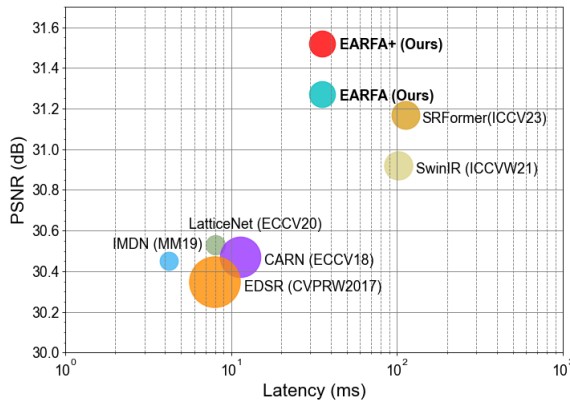

**Figure 1: Comparison of the tradeoff between SR results and model efficiency on Manga109 [4] with SR×4. The diameter of each circle denotes the Multi-Adds [5] of the corresponding model. Our EARFA achieves the best SR performance while keeping fast reasoning speed.**

Convolutional neural network (CNN)-based models are the most common methods for ESISR [7–11]. This kind of methods is primarily implemented by conventional convolutional operations, and generally works with relatively high inference efficiency. However, due to limited receptive field and inefficient feature utilization, the performance of such models is usually unsatisfactory. Another type of models are built upon more advanced Transformer architectures [12] and greatly push the performance margin of ESISR, such as SwinIR [13] and SRFormer [14]. A remarkable feature of these methods is that they can achieve better SR results with fewer model parameters. But the multi-head self-attention (MSA) inherent in the Transformer architecture involves a large number of complex calculations, essentially leading to inefficient SR inference.

Subsequently, researchers began to seek a better balance between model performance and reasoning efficiency through improving the representational capacity of CNN-based models and accelerating mapping inference of Transformer-based models. For example, most earlier CNN-based models focused on designing more compact network architectures to increase model representation and decrease model parameters, such as global residual structure [8], feature pyramid network [15], recursive networks based on parameter sharing [16, 17], information distillation network [18], persistent memory network [19] etc. Another way to improve the representational capacity of SISR models is to improve feature utilization through attention mechanisms, including channel attention-based RCAN [20], spatial attention-based CSFM [21] and SAN [22], layer-wsie attention-based HAN [23], attention cube-based A-CubeNet

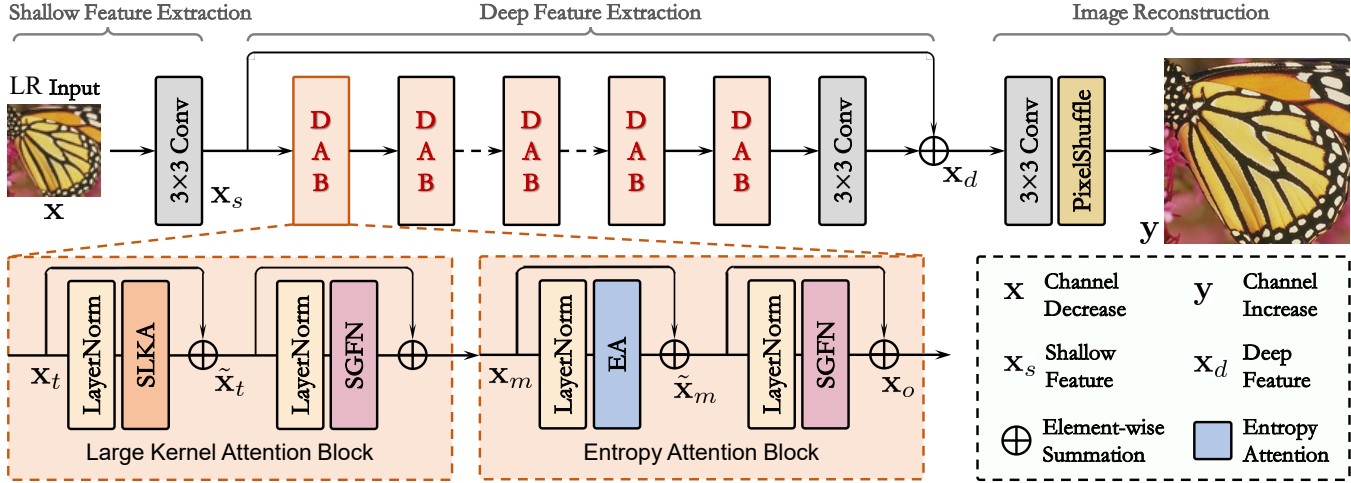

Figure 2: The overall structure of our EARFA. DAB constitutes the basic module for nonlinear inference, and LKAB and EAB are the building components of DAB that integrate SLKA and EA, respectively.

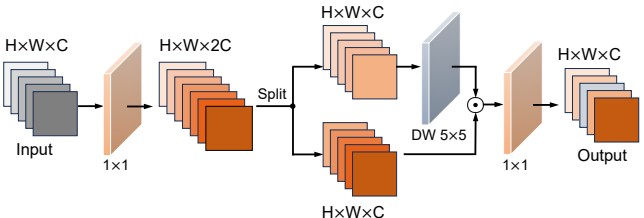

Figure 3: The architecture of SGFN. 1 × 1 denotes a convolutional layer with a kernel size of 1×1, and Split refers to splitting input features into two parts along the channel dimension, while DW5 × 5 denotes a depth-wise convolution with a 5×5 kernel size.

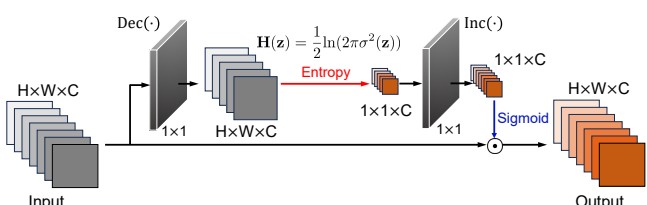

Figure 4: The network architecture of EA, where 1 × 1 denotes the convolutional layer with a kernel size of 1×1. **Entropy** signifies the computation of the differential entropy [6] for channel-wise features, and **Sigmoid** denotes the sigmoid function for weight normalization.

[24], nonlocal attention-based NLSN [25], as well as directional variance attention-based DiVANet [26]. These variant models based on attention mechanisms can improve SR performance to a certain extent, but they are often accompanied by an increase in computational complexity. Besides, most of these models show evident performance improvement for large-scale models, but their effectiveness for ESISR tasks is not significant. For Transformer-based

models, Zhang *et al.* [27] employed shift convolution (shift-conv) to effectively extract the image local structural information while maintaining the same level of complexity as 1×1 convolution. Zhou *et al.* [14] proposed the permuted self-attention (PSA) that strikes a proper balance between channel and spatial information. Although these methods appropriately shrunk the parameter scale of Transformer-based SISR models, the problem of inference efficiency still looms prominent.

In terms of SISR tasks, an important reason why Transformer-based models typically perform better than CNN-based models is that MSA achieves non-local information perception and expands the effective receptive field of the models at the cost of higher computational overhead. Therefore, in this work, we propose a novel **E**ntropy **A**ttention and **R**eceptive **F**ield **A**ugmentation (**EARFA**) model for ESISR from the perspective of reducing the computational overhead and increasing the effective receptive field of the model. It consists of an **E**ntropy **A**ttention (**EA**) mechanism for efficient utilization of intermediate features and a **S**hifting **L**arge **K**ernel **A**ttention (**SLKA**) for augmenting effective receptive field of the model and diversity of hierarchical features. **EA** is introduced into the model to elevate the entropy of intermediate features conditioned on a Gaussian distribution, and thus increase the input information for subsequent inference. Specifically, it computes the differential entropy [6] for channel-wise features, which is used to measure the information amount in randomly distributed data. And the attention weights are obtained by driving the features approaching to a Gaussian distribution. **SLKA** is an improved version of lager kernel attention (LKA) [28] aimed at further augmenting the effective receptive field of the model with negligible overhead. This is implemented by simply shifting partial channels of a intermediate feature [29]. It is worth noting that both our EA and SLKA do not involve complex calculations, which avoids significant delays in the inference process of the proposed **EARFA** model. Fig.1 illustrates the comparison of the tradeoff between model performance and efficiency. As can be seen, our **EARFA** model achieves better SR results with less inference delay compared to Transformer-based

SwinIR [13] and SRFormer [14], despite maintaining slightly more model parameters.

In summary, the primary contributions of this work are three-fold as follows:

- From the perspective of information theory, we introduce a novel **EARFA** model for ESISR tasks. It achieves superior SR performance to most existing models and boasts faster inference than advanced Transformer-based models.

- A new attention mechanism (i.e., **EA**) based on differential entropy has been crafted as a new criterion for evaluating the significance of channel-wise features. Unlike traditional attention mechanisms modeled on biological attention mechanisms in neuroscience, our **EA** is motivated by information theory to improve the information degree of hierarchical features via increasing the differential entropy of intermediate features.

- We propose to augment the effective receptive field of the model with a simple yet efficient variant of LKA [28], which replaces the point-wise convolution with a shifting convolution. The substitution can not only eliminate the computational overhead of the point-level convolution, but also increase the feature diversity and model receptive field.

## 2 RELATED WORK

### 2.1 Efficient SISR based on Deep Learning

Initially, the first wave of efficient SISR methods [7, 8, 30] based on deep learning predominantly utilized interpolation algorithms for preprocessing. They employed these algorithms to reconstruct low-resolution images into high-resolution ones before further enhancing them using deep learning models to achieve the final result. As the input image size matched the original dimensions, these methods often incurred higher latency. To mitigate latency issues, subsequent efficient SISR [9, 20] approaches began directly feeding low-resolution images into the model, completing image reconstruction at the model's end to generate high-resolution images. In recent years, the emergence of Transformer-based efficient SISR methods has delivered exceptional performance. These models typically exhibit small parameter size while showcasing remarkable efficiency. Nonetheless, the self-attention mechanism in Transformer is computationally complex, which As a result, the latency of this type of method is high.

Some of the aforementioned methods exhibit inadequate reconstruction performance, while others suffer from high latency. As a result, they fail to strike a balance between reconstruction quality and inference speed, ultimately lacking sufficient efficiency.

### 2.2 Attention Mechanisms for Efficient SISR

Since attention mechanisms [31] were introduced, they have been widely adopted across various networks to bolster model performance. In the domain of ESISR, early efforts were concentrated on channel and spatial attentions, which involve manipulating feature maps through pooling and activation of high-frequency areas, respectively. Recent advancements have seen LKA [28] and MSA significantly improving ESISR outcomes. LKA [28] extends a

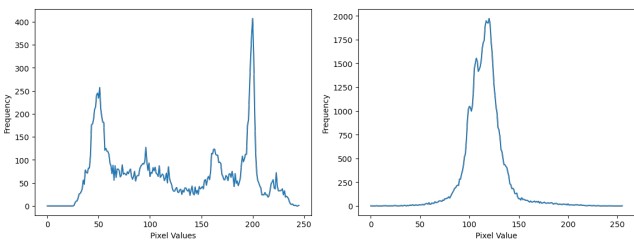

**Figure 5: Pixel distribution of intermediate features. The left illustrates the pixel distribution of the input feature, while the right shows the pixel distribution of the features after adjustment of approaching to the Gaussian distribution.**

model's receptive-field by merging dilated and depth-wise separable convolutions, minimizing parameter count and computational load. Whereas MSA is a staple in Transformer architectures, is celebrated for its broad utility.

ESISR struggles to preserve extensive information for reconstruction owing to their limited parameter count. Nevertheless, the conventional attention mechanism has not taken this aspect into account, resulting in average performance in ESISR.

### 2.3 Receptive Field Augmentation

Due to the limited depth of ESISR models, the effective receptive field of these models [7, 32] are typically deficient, leading to a limited model representational capacity. In order to enhance the reconstruction performance of ESISR models, it is necessary to offer the network with a larger receptive field without compromising the speed of model inference. Expanding the model's receptive field commonly involves utilizing convolutional layers with larger kernels [33], which is a straightforward and efficient approach. In transformer-based method [13, 14], enlarging the window size during self-attention calculation within MSA can also increase the receptive field.

While the aforementioned approaches can enhance the model's receptive field, they introduce additional computational complexity or parameters, rendering them unsuitable for ESISR.

## 3 METHODOLOGY

In this section, we first introduce the overall structure of the proposed **EARFA**, and then demonstrate the principles of **EA** and **SLKA**, respectively.

### 3.1 Overall Architecture

The overall structure of EARFA is shown in Fig. 2, which is mainly divided into three parts: shallow feature extraction, deep feature extraction and image reconstruction. We first extract shallow features with a simple 3×3 convolutional layer:

$$\mathbf{x}_s = \mathrm{SF}(\mathbf{x}), \tag{1}$$

where $\mathbf{x}$ is the input image, and $\mathrm{SF}(\cdot)$ represents the shallow feature extraction performed using the 3×3 convolution. The obtained shallow feature is denoted as $\mathbf{x}_s$.

Next, we cascade multiple dual-attention blocks (DAB) to form the whole nonlinear mapping. As shonw in Fig. 2, each DAB consists

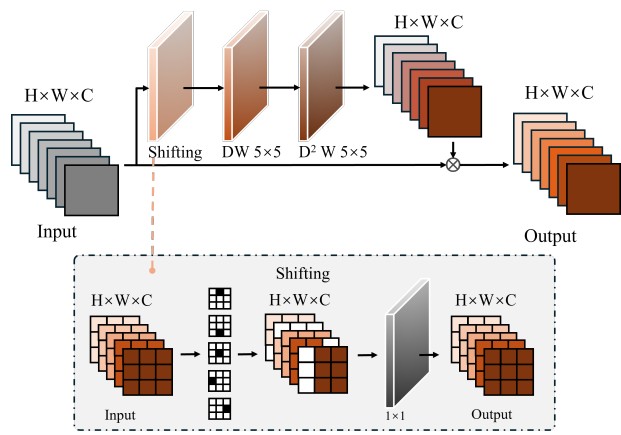

**Figure 6: The network architecture of SLKA.** Shifting **denotes shifting convolution with a kernel size of 1×1, and** DW5 × 5 **represents the depth-wise convolution with a kernel size of 5×5, and** $D^2W5 \times 5$ **stands for the depth-wise convolution with a dilation rate of 3 and a kernel size of 5×5.**

of two components: a large kernel attention block (LKAB) and an entropy attention block (EAB). Given a temporary feature $\mathbf{x}_t$, the mapping process of DAB can be represented as:

$$
\begin{aligned}
\tilde{\mathbf{x}}_t &= \text{SLKA}(\text{LayerNorm}(\mathbf{x}_t)) + \mathbf{x}_t, \\
\mathbf{x}_m &= \text{SGFN}(\text{LayerNorm}(\tilde{\mathbf{x}}_t)) + \tilde{\mathbf{x}}_t, \\
\tilde{\mathbf{x}}_m &= \text{EA}(\text{LayerNorm}(\mathbf{x}_m)) + \mathbf{x}_m, \\
\mathbf{x}_o &= \text{SGFN}(\text{LayerNorm}(\tilde{\mathbf{x}}_m)) + \tilde{\mathbf{x}}_m
\end{aligned}
\tag{2}
$$

where LayerNorm($\cdot$) denotes the layer normalization, and $\tilde{\mathbf{x}}_t$ is the result of our **SLKA** enhanced via channel shifting, which possesses augmented receptive field and feature diversity. $\tilde{\mathbf{x}}_m$ represents the result of our **EA** attention that contains more informative channel-wise features. The spatial-gate feed-forward network (SGFN) [34] is adopted to aggregate the channel-wise spatial information of intermediate features that contain augmented receptive field and differential entropy. The structure of SGFN used in this work is illustrated in Fig. 3. It is worth noting that for receptive field and entropy enhancement, SGFN adopts a two-stage mode to boost the features progressively, as shown in Fig. 2 and Eqn. 2.

In the image reconstruction section, we utilize a convolutional layer with a kernel size of 3 × 3 followd by a PixelShuffle layer to recover HR image, which can be represented as:

$$
\mathbf{y} = \text{PixelShuffle}(\text{Conv}_{3\times3}(\mathbf{x}_d)),
\tag{3}
$$

where $\mathbf{x}_d$ denotes the deep feature obtained by the stacked DABs, and $\text{Conv}_{3\times3}(\cdot)$ indicates the convolutional layer with kernel size of 3 × 3. PixelShuffle($\cdot$) is used to upscale the final feature and shrink the number of feature channels to 3, and $\mathbf{y}$ is the HR output.

### 3.2 Entropy Attention

Traditional channel-wise attention mechanisms utilize pooling to aggregate information from each channel. The advantage of this approach is the simple acquisition of weights for each channel-wise feature, thereby enhancing model efficiency. However, ESISR

models typically maintain a small parameter scale and limited information capacity. Thus, for ESISR tasks, we aim at measuring the weight of each feature map with high efficiency while also providing more information for image reconstruction. To this end, we compute the information entropy [6] of each feature map as an alternative to pooling operations. Formally, information entropy [6] can be computed as:

$$
\mathbf{H}(\mathbf{z}) = -\sum_k p(z_k) \cdot \log\big[p(z_k)\big],
\tag{4}
$$

where $\mathbf{z}$ represents a set of data following a random discrete distribution, and $p(z_k)$ denote the probability distribution of $z_k$, where $z_k$ is the value in $\mathbf{z}$ that actually participates in the calculation. $\mathbf{H}(\mathbf{z})$ stands for the information entropy [6] of dataset $\mathbf{z}$.

However, the calculation of information entropy in Eqn. (4) is only applicable to data following discrete random distributions, and cannot be used to compute the information entropy of continuous random variables. This is the case for intermediate features of deep neural networks. Therefore, we opt to calculate the differential entropy [6], which is suitable for data obeying continuous random distributions. The expression for the differential entropy [6] can be formulated as:

$$
\mathbf{H}(\mathbf{z}) = -\int_{z_k \in \mathcal{Z}} p(z_k) \cdot \log\big[p(z_k)\big] dz_k,
\tag{5}
$$

where $\mathbf{z}$ represents a set of data following a continuous random distribution $\mathcal{Z}$, and $p(\cdot)$ is the probability density. $z_k$ is a continuous variables in $\mathbf{z}$ that actually participates in the calculation. $\mathbf{H}(\mathbf{z})$ is the differential entropy [6] of dataset $\mathbf{z}$. Based on the formula for differential entropy [6], we can calculate it for each channel-wise feature and design a novel method for feature enhancement with entropy attention.

However, typical differential entropy involves differentiation and probability density distribution estimation for continuous data, which is intractable and time-consuming. This limitation indicates that the conventional method of calculating differential entropy is not suitable for scenarios of computing the differential entropy of hierarchical features in neural networks. Therefore, we turn to process the features to align them as closely as possible with a Gaussian distribution. As demonstrated in Fig. 4, the computation of the differential entropy conditioned on a Gaussian distribution can be written as:

$$
\mathbf{H}(\mathbf{z}) = \frac{1}{2} \ln \big(2\pi\sigma^2(\mathbf{z})\big).
\tag{6}
$$

Here, $\sigma(\cdot)$ is the standard deviation, and $\ln(\cdot)$ signifies the napierian logarithm, and $\mathbf{H}(\mathbf{z})$ represents the differential entropy [6] of the feature map conditioned on the Gaussian distribution. The calculation of $\mathbf{H}(\mathbf{z})$ is solely dependent on the variance, resulting more efficient nonlinear inference. As illustrated in Table 4, compared to traditional methods of computing the differential entropy, the computation manner used in this work introduces almost no inference delay under the same running environment.

To better approximate a Gaussian distribution, we apply layer normalization to preprocess the features before EA. Within EA, we utilize a convolutional layer with a kernel size of 1 × 1 to reduce the number of feature channels while further refining the features. As shown in Fig. 5, we visualize the distribution of features before

Table 1: Ablation investigation of different configurations of EARFA with SR×4. We compared the results of LKA [28] and SLKA, SE and EA respectively (PSNR (dB)/SSIM).

| Model | SLKA [Ours] | EA [Ours] | LKA [28] | SE [39] | Set5 [35] PSNR | SSIM | Set14 [36] PSNR | SSIM | BSD100 [37] PSNR | SSIM | Urban100 [38] PSNR | SSIM | Manga109 [4] PSNR | SSIM |
|---|---|---|---|---|---|---|---|---|---|---|---|---|---|---|
| EARFA | ✗ | ✗ | ✗ | ✗ | 32.17 | 0.8954 | 28.64 | 0.7831 | 27.61 | 0.7374 | 26.17 | 0.7883 | 30.64 | 0.9103 |
|  | ✔ | ✗ | ✗ | ✗ | 32.49 | 0.8988 | 28.82 | 0.7875 | 27.74 | 0.7425 | 26.59 | 0.8022 | 31.11 | 0.9164 |
|  | ✗ | ✔ | ✗ | ✗ | 32.43 | 0.8978 | 28.72 | 0.7844 | 27.65 | 0.7388 | 26.40 | 0.7934 | 30.91 | 0.9134 |
|  | ✗ | ✔ | ✔ | ✗ | 32.56 | **0.8998** | 28.84 | 0.7878 | 27.75 | 0.7429 | 26.64 | 0.8035 | 31.23 | 0.9176 |
|  | ✔ | ✗ | ✗ | ✔ | 32.55 | 0.8991 | 28.83 | 0.7876 | 27.74 | 0.7427 | 26.62 | 0.8030 | 31.12 | 0.9167 |
|  | ✔ | ✔ | ✗ | ✗ | **32.58** | 0.8995 | **28.86** | **0.7879** | **27.76** | **0.7431** | **26.70** | **0.8044** | **31.27** | **0.9177** |

Table 2: Quantitative comparison for efficient SISR on benchmark datasets (PSNR (dB) / SSIM). MultAdds [5] and Latency are computed via upscaling an image to 1280×720 resolution on a NVIDIA RTX 4090 GPU. "+" indicates the model is trained on DF2K dataset. The best and second best results are marked in red and blue, respectively.

| Model | Annual | Scale | Params (K) | Multi-Adds (G) | Latency (ms) | Set5 [35] PSNR | SSIM | Set14 [36] PSNR | SSIM | BSD100 [37] PSNR | SSIM | Urban100 [38] PSNR | SSIM | Manga109 [4] PSNR | SSIM |
|---|---|---|---|---|---|---|---|---|---|---|---|---|---|---|---|
| EDSR-baseline [9] | CVPRW17 | ×2 | 1370 | 316.3 | 26.33 | 37.99 | 0.9604 | 33.57 | 0.9175 | 32.16 | 0.8994 | 31.98 | 0.9272 | 38.54 | 0.9769 |
| CARN [5] | ECCV18 | | 1592 | 222.8 | 36.98 | 37.76 | 0.959 | 33.52 | 0.9166 | 32.09 | 0.8978 | 31.92 | 0.9256 | 38.36 | 0.9765 |
| IMDN [40] | ACM'MM19 | | 694 | 158.8 | 18.17 | 38.00 | 0.9605 | 33.63 | 0.9177 | 32.19 | 0.8996 | 32.17 | 0.9283 | 38.88 | 0.9774 |
| LatticeNet [41] | ECCV20 | | 756 | 169.5 | 23.04 | 38.06 | 0.9607 | 33.70 | 0.9187 | 32.20 | 0.8999 | 32.25 | 0.9288 | 38.94 | 0.9774 |
| ESRT [42] | CVPRW22 | | - | - | OOM | - | - | - | - | - | - | - | - | - | - |
| SwinIR [13] | ICCVW21 | | 910 | 252.9 | 958.14 | 38.14 | 0.9611 | 33.86 | 0.9206 | 32.31 | 0.9012 | 32.76 | 0.934 | 39.12 | 0.9783 |
| SRFormer [14] | ICCV23 | | 853 | 236.2 | 1015.23 | 38.23 | 0.9613 | 33.94 | 0.9209 | 32.36 | 0.9019 | 32.91 | 0.9353 | 39.28 | 0.9785 |
| **EARFA** | **Ours** | | 1026 | 229.0 | 182.68 | 38.24 | 0.9614 | 33.98 | 0.9212 | 32.36 | 0.9022 | 32.98 | 0.9359 | 39.36 | 0.9785 |
| **EARFA+** | **Ours** | | 1026 | 229.0 | 182.68 | 38.27 | 0.9616 | 34.14 | 0.9229 | 32.41 | 0.9028 | 33.20 | 0.9376 | 39.63 | 0.9790 |
| EDSR-baseline [9] | CVPRW17 | ×3 | 1555 | 160.2 | 9.58 | 34.37 | 0.927 | 30.28 | 0.8417 | 29.09 | 0.8052 | 28.15 | 0.8527 | 33.45 | 0.9439 |
| CARN [5] | ECCV18 | | 1592 | 118.8 | 14.79 | 34.29 | 0.9255 | 30.29 | 0.8407 | 29.06 | 0.8034 | 28.06 | 0.8493 | 33.5 | 0.944 |
| IMDN [40] | ACM'MM19 | | 703 | 71.5 | 5.22 | 34.36 | 0.927 | 30.32 | 0.8417 | 29.09 | 0.8046 | 28.17 | 0.8519 | 33.61 | 0.9445 |
| LatticeNet [41] | ECCV20 | | 765 | 76.3 | 8.36 | 34.40 | 0.9272 | 30.32 | 0.8416 | 29.10 | 0.8049 | 28.19 | 0.8513 | 33.63 | 0.9442 |
| ESRT [42] | CVPRW22 | | 770 | 96.4 | OOM | 34.42 | 0.9268 | 30.43 | 0.8433 | 29.15 | 0.8063 | 28.46 | 0.8574 | 33.95 | 0.9455 |
| SwinIR [13] | ICCVW21 | | 918 | 114.5 | 389.89 | 34.62 | 0.9289 | 30.54 | 0.8463 | 29.20 | 0.8082 | 28.66 | 0.8624 | 33.98 | 0.9478 |
| SRFormer [14] | ICCV23 | | 861 | 104.8 | 224.54 | 34.67 | 0.9296 | 30.57 | 0.8469 | 29.26 | 0.8099 | 28.81 | 0.8655 | 34.19 | 0.9489 |
| **EARFA** | **Ours** | | 1034 | 102.4 | 65.40 | 34.73 | 0.9297 | 30.61 | 0.8471 | 29.29 | 0.8103 | 28.87 | 0.8663 | 34.38 | 0.9494 |
| **EARFA+** | **Ours** | | 1034 | 102.4 | 65.40 | 34.77 | 0.9302 | 30.68 | 0.8486 | 29.34 | 0.8114 | 29.04 | 0.8695 | 34.69 | 0.9507 |
| EDSR-baseline [9] | CVPRW17 | ×4 | 1518 | 114 | 7.98 | 32.09 | 0.8938 | 28.58 | 0.7813 | 27.57 | 0.7357 | 26.04 | 0.7849 | 30.35 | 0.9067 |
| CARN [5] | ECCV18 | | 1592 | 90.9 | 11.32 | 32.13 | 0.8937 | 28.60 | 0.7806 | 27.58 | 0.7349 | 26.07 | 0.7837 | 30.47 | 0.9084 |
| IMDN [40] | ACM'MM19 | | 715 | 40.9 | 4.22 | 32.21 | 0.8948 | 28.58 | 0.7811 | 27.56 | 0.7353 | 26.04 | 0.7838 | 30.45 | 0.9075 |
| LatticeNet [41] | ECCV20 | | 777 | 43.6 | 8.05 | 32.18 | 0.8943 | 28.61 | 0.7812 | 27.57 | 0.7355 | 26.14 | 0.7844 | 30.53 | 0.9075 |
| ESRT [42] | CVPRW22 | | 751 | 67.7 | OOM | 32.19 | 0.8947 | 28.69 | 0.7833 | 27.69 | 0.7379 | 26.39 | 0.7962 | 30.75 | 0.9100 |
| SwinIR [13] | ICCVW21 | | 930 | 65.2 | 101.68 | 32.44 | 0.8976 | 28.77 | 0.7858 | 27.69 | 0.7406 | 26.47 | 0.798 | 30.92 | 0.9151 |
| SRFormer [14] | ICCV23 | | 873 | 62.8 | 112.15 | 32.51 | 0.8988 | 28.82 | 0.7872 | 27.73 | 0.7422 | 26.67 | 0.8032 | 31.17 | 0.9165 |
| **EARFA** | **Ours** | | 1045 | 58.4 | 35.40 | 32.58 | 0.8995 | 28.86 | 0.7879 | 27.75 | 0.7431 | 26.70 | 0.8044 | 31.27 | 0.9177 |
| **EARFA+** | **Ours** | | 1045 | 58.4 | 35.40 | 32.62 | 0.9003 | 28.94 | 0.7898 | 27.81 | 0.7444 | 26.86 | 0.8081 | 31.52 | 0.9197 |

computing the differential entropy [6] and observe that the features follow an approximate Gaussian distribution.

As shown in Fig. 4, the computation process of our **EA** attention can be formulated as:

$$\mathbf{y}_t = \text{Inc}(\text{Sig}(\text{Cde}(\text{Dec}(\mathbf{x}_t)))) \odot \mathbf{x}_t \tag{7}$$

Where $\mathbf{x}_t$ represents the input to **EA**. $\text{Dec}(\cdot)$ denotes the operation of using a convolutional layer with a kernel size of 1×1 to reduce the number of channels while refining features, and $\text{Cde}(\cdot)$ signifies computing the differential entropy of each feature map. $\text{Sig}(\cdot)$ is

the sigmoid function, and $\text{Inc}(\cdot)$ stands for restoring the number of feature channels, and $\odot$ indicates element-wise multiplication.

## 3.3 Shifting Large Kernel Attention

LKA [28] mainly enhances the receptive field of the model by utilizing dilated convolution and depth-wise convolution, addressing the limited receptive field issue in ESISR methods and thereby improving ESISR performance. However, in LKA [28], it is not necessarily better for the kernel size of depth-wise convolutions and the dilation rate of dilated convolutions to be larger. Therefore, we need to

Table 3: Quantitative comparison for super lightweight SISR on benchmark datasets (PSNR (dB) / SSIM). MultAdds [5] and Latency are computed via upscaling an image to 1280×720 resolution on a NVIDIA RTX 4090 GPU. "+" indicates the model is trained on DF2K dataset. The best and second best results are marked in red and blue, respectively.

| Model | Annual | Scale | Params (K) | Multi-Adds (G) | Latency (ms) | Set5 [35] PSNR | Set5 [35] SSIM | Set14 [36] PSNR | Set14 [36] SSIM | BSD100 [37] PSNR | BSD100 [37] SSIM | Urban100 [38] PSNR | Urban100 [38] SSIM | Manga109 [4] PSNR | Manga109 [4] SSIM |
|---|---|---|---|---|---|---|---|---|---|---|---|---|---|---|---|
| SRCNN [7] | TPAMI15 | | 57 | 52.7 | 6.85 | 36.66 | 0.9545 | 32.42 | 0.9063 | 31.36 | 0.8879 | 29.50 | 0.8946 | 35.60 | 0.9663 |
| VDSR [8] | CVPR16 | | 665 | 612.6 | 32.70 | 37.53 | 0.9587 | 33.03 | 0.9124 | 31.90 | 0.8960 | 30.76 | 0.9140 | 37.22 | 0.9729 |
| DRRN [17] | CVPR17 | | 297 | 6796 | 250.28 | 37.74 | 0.9591 | 33.23 | 0.9136 | 32.05 | 0.8973 | 31.23 | 0.9188 | 37.88 | 0.9749 |
| IDN [18] | CVPR18 | | 553 | 127 | 16.1 | 37.83 | 0.9600 | 33.30 | 0.9148 | 32.08 | 0.8985 | 31.27 | 0.9196 | 38.01 | 0.9749 |
| PAN [32] | ECCVW20 | ×2 | 261 | 70.5 | 12.76 | 38.00 | 0.9605 | 33.59 | 0.9181 | 32.18 | 0.8997 | 32.01 | 0.9273 | 38.70 | 0.9773 |
| ShuffleMixer [43] | NIPS22 | | 394 | 91 | 36.46 | 38.01 | 0.9606 | 33.63 | 0.9180 | 32.17 | 0.8995 | 31.89 | 0.9257 | 38.83 | 0.9774 |
| SAFMN [10] | ICCV23 | | 228 | 52 | 15.83 | 38.00 | 0.9605 | 33.54 | 0.9177 | 32.16 | 0.8995 | 31.84 | 0.9256 | 38.71 | 0.9771 |
| **EARFA-light** | **Ours** | | 199 | 44.05 | 63.79 | 38.08 | 0.9608 | 33.64 | 0.9188 | 32.23 | 0.9004 | 32.27 | 0.9297 | 38.85 | 0.9774 |
| **EARFA-light+** | **Ours** | | 199 | 44.05 | 63.79 | 38.05 | 0.9608 | 33.65 | 0.9188 | 32.23 | 0.9005 | 32.28 | 0.9298 | 39.10 | 0.9781 |
| SRCNN [7] | TPAMI15 | | 57 | 52.7 | 6.85 | 32.75 | 0.9090 | 29.28 | 0.8209 | 28.41 | 0.7863 | 26.24 | 0.7989 | 30.48 | 0.9117 |
| VDSR [8] | CVPR16 | | 665 | 612.6 | 32.70 | 33.66 | 0.9213 | 29.77 | 0.8314 | 28.82 | 0.7976 | 27.14 | 0.8279 | 32.01 | 0.9310 |
| DRRN [17] | CVPR17 | | 297 | 6796 | 250.28 | 34.03 | 0.9244 | 29.96 | 0.8349 | 28.95 | 0.8004 | 27.53 | 0.8378 | 32.71 | 0.9379 |
| IDN [18] | CVPR18 | | 553 | 57 | 6.5 | 34.11 | 0.9253 | 29.99 | 0.8354 | 28.95 | 0.8013 | 27.42 | 0.8359 | 32.71 | 0.9381 |
| PAN [32] | ECCVW20 | ×3 | 261 | 39.0 | 7.42 | 34.40 | 0.9271 | 30.36 | 0.8423 | 29.11 | 0.8050 | 28.11 | 0.8511 | 33.61 | 0.9448 |
| ShuffleMixer [43] | NIPS22 | | 415 | 43 | 13.00 | 34.40 | 0.9272 | 30.37 | 0.8423 | 29.12 | 0.8051 | 28.08 | 0.8498 | 33.69 | 0.9448 |
| SAFMN [10] | ICCV23 | | 233 | 23 | 6.01 | 34.34 | 0.9267 | 30.33 | /0.8418 | 29.08 | 0.8048 | 27.95 | 0.8474 | 33.52 | 0.9437 |
| **EARFA-light** | **Ours** | | 203 | 20.03 | 25.72 | 34.47 | 0.9276 | 30.40 | 0.8430 | 29.15 | 0.8064 | 28.26 | 0.8542 | 33.89 | 0.9462 |
| **EARFA-light+** | **Ours** | | 203 | 20.03 | 25.72 | 34.48 | 0.9280 | 30.44 | 0.8438 | 29.16 | 0.8067 | 28.29 | 0.8549 | 33.94 | 0.9466 |
| SRCNN [7] | TPAMI15 | | 57 | 52 | 6.85 | 30.48 | 0.8628 | 27.49 | 0.7503 | 26.90 | 0.7101 | 24.52 | 0.7221 | 27.58 | 0.8555 |
| VDSR [8] | CVPR16 | | 665 | 612.6 | 32.70 | 31.35 | 0.8838 | 28.01 | 0.7674 | 27.29 | 0.7251 | 25.18 | 0.7524 | 28.83 | 0.8809 |
| DRRN [17] | CVPR17 | | 297 | 6796 | 250.28 | 31.68 | 0.8888 | 28.21 | 0.7720 | 27.38 | 0.7284 | 25.44 | 0.7638 | 29.45 | 0.8946 |
| IDN [18] | CVPR18 | | 553 | 32.3 | 3.23 | 31.82 | 0.8903 | 28.25 | 0.7730 | 27.41 | 0.7297 | 25.41 | 0.7632 | 29.41 | 0.8942 |
| PAN [32] | ECCVW20 | ×4 | 272 | 28.2 | 6.63 | 32.13 | 0.8948 | 28.61 | 0.7822 | 27.59 | 0.7363 | 26.11 | 0.7854 | 30.51 | 0.9095 |
| ShuffleMixer [43] | NIPS22 | | 411 | 28 | 9.19 | 32.21 | 0.8953 | 28.66 | 0.7827 | 27.61 | 0.7366 | 26.08 | 0.7835 | 30.65 | 0.9093 |
| SAFMN [10] | ICCV23 | | 240 | 14 | 5.80 | 32.18 | 0.8948 | 28.60 | 0.7813 | 27.58 | 0.7359 | 25.97 | 0.7809 | 30.43 | 0.9063 |
| **EARFA-light** | **Ours** | | 209 | 11.61 | 14.82 | 32.29 | 0.8963 | 28.63 | 0.7828 | 27.61 | 0.7380 | 26.22 | 0.7898 | 30.62 | 0.9105 |
| **EARFA-light+** | **Ours** | | 209 | 11.61 | 14.82 | 32.33 | 0.8964 | 28.68 | 0.7832 | 27.64 | 0.7382 | 26.20 | 0.7889 | 30.75 | 0.9115 |

employ other methods to further enhance the receptive field size and consequently bolster the model's expressive power.

To further expand the model's receptive field and enhance its expressive capability, we introduce SLKA that replaces the point-wise convolutions in LKA [28] with shifting convolutions using a kernel size of $1 \times 1$. As illustrated in Fig. 6, the shifting convolution layer divides the input features into five groups along the channel dimension, keeping one group unchanged while shifting the other four groups one pixel in the directions of top, bottom, left and right. This effectively allows each 1×1 convolution kernel to simultaneously process features from five pixels in the top, bottom, left, right, and center positions, resulting in a receptive field size five times larger than that of a standard 1×1 convolution. As depicted in Fig. 6, our SLKA can be represented as:

$$\mathbf{y}_t = D^2W(DW(SC(\mathbf{x}_t))) \odot x \tag{8}$$

Where $\mathbf{x}_t$ represents the input features to SLKA, $SC(\cdot)$ denotes the shifting convolution layer with a kernel size of 1×1, $DW(\cdot)$ represents the depth-wise convolution layer, $D^2W(\cdot)$ signifies the dilated depth-wise convolution layer. The symbol $\odot$ indicates element-wise multiplication. We will compare the effectiveness of LKA [28] and SLKA in our ablation study.

## 4 EXPERIMENTS

In this part, we will describe our experimental results. We first present the feasibility of the proposed method through ablation studies, and then compare it with other advanced SISR methods w.r.t both ESISR tasks and ESISR-light tasks to showcase the superiority of our **EARFA**, where the latter corresponds to super lightweight SISR models.

### 4.1 Experimental Settings

**Datasets and Evaluation.** Following the general conventions in SR community, we first chose DIV2K [44] as the training dataset of our **EARFA**. It is one of the most commonly-used dataset for training SISR models that contains 800 high-definition images. For further validation and evaluation, we also trained our **EARFA** with a larger dataset DF2K (DIV2K [44] + Flick2K [45]), which is denoted as **EARFA+** or **EARFA-light+**. To inspect the generalization capability of these models, we selected Set5 [35], Set14 [36], BSD100 [37], Urban100 [38], and Manga109 [46] as our test datasets, all of which are popular benchmarks in the field. The experimental results are evaluated in terms of PSNR (dB) and SSIM [47], which are calculated on the Y channel from the YCbCr color space.

**Implementation Details.** Our **EARFA** consists of 12 DABs, with the channel compression ratio in **EA** set to 8. This indicates that the number of channels is reduced to 1/8 of the input features during

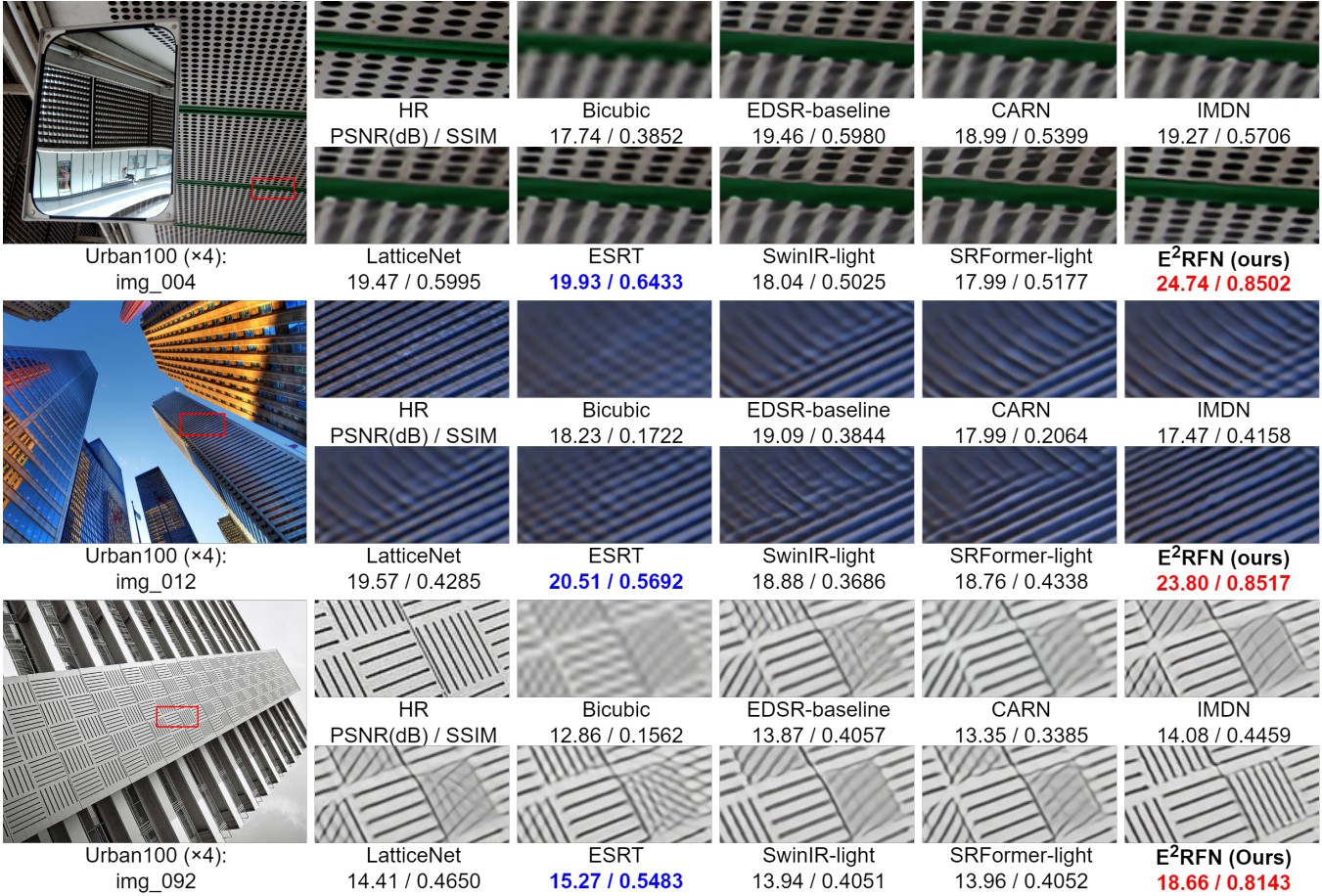

**Figure 7: Visual comparison between the proposed models and other advanced SISR methods on Urban100 [38] with SR×4. The best and second best results are marked in red and blue, respectively.**

the processing in **EA**. In **SLKA**, the kernel sizes of shifting convolution, depth-wise convolution, and dilated depth-wise convolution are 1×1, 5×5, and 7×7, respectively. The dilation rate of the dilated depth-wise convolution is set to 3. The kernel size of the depth-wise convolution in SGFN [34] is 5×5.

For **EARFA-light**, 8 DABs are included in the nonlinear mapping with the settings of **EA** unchanged. The kernel sizes for shifting convolution, depth-wise convolution and dilated depth-wise convolution are 1×1, 3×3 and 5×5, respectively. The dilation rate of the dilated depth-wise convolution remains to be 3. The kernel size of the depth-wise convolution in SGFN [34] is 3×3.

**Training Settings.** During training, **EARFA** takes input image patches of size 64×64 with a batch size of 64. It adopts the Adam [48] optimizer with $\beta_1 = 0.9$ and $\beta_2 = 0.99$ to minimize the L1 loss. The whole training process undergoes 500K iterations. The initial learning is set to $5 \times 10^{-4}$ and is halved at 250k, 400k, 450k and 475k iterations. Additionally, standard data augmentation techniques such as rotation and horizontal flipping are used for training more robust models. For **EARFA-light**, the initial learning rate is set to $1 \times 10^{-3}$ and also halved at the same steps as **EARFA**. Other

configurations for **EARFA-light** remain the same. The proposed models are implemented using PyTorch and trained on a NVIDIA GeForce RTX 2080Ti GPU.

## 4.2 Ablation Study

**Comparison of Attention.** As presented in Table 1, the baseline model without attention achieves PSNR values of 26.17 dB and 30.64 dB on Urban100 [38] and Manga109 [4], respectively. Deploying SLKA on this basic model results in gains of **0.42 dB** on Urban100 [38] and **0.47dB** on Manga109 [4], while deploying EA leads to gains of **0.23dB** on Urban100 [38] and **0.27dB** on Manga109 [4]. Although the improvements of **EA** are less than **SLKA**, it only contains ∼2k parameters and can be used as an efficient plug-and-play module for typical low-level vision tasks.

On the other hand, compared to the model with EA and LKA [28], it is evident that deploying models with EA and SLKA yields more higher PSNR values by **0.06 dB** and **0.04 dB** on Urban100 [38] and Manga109 [4], respectively. Similarly, compared to the model with SE and SLKA, the PSNR values are higher by **0.08 dB** and **0.15 dB** on Urban100 [38] and Manga109 [4], respectively. These

experimental results validate the feasibility and robustness of the proposed structural components.

**Table 4:** Tradition **represents the traditional way to compute differential entropy, and** Gaussian **here denotes computing it conditioned on a Gaussian distribution.** Avg.Time **is calculated for upsaling an image from 320×180 to 1280×720 resolution on NVIDIA RTX 4090 GPU.**

| Batch Size | Method | #Avg. Time |
|---|---|---|
| 8 | Tradition [6] | 4.79 ms |
| | Gaussian | 0.81 ms |
| 16 | Tradition [6] | 9.79 ms |
| | Gaussian | 1.19 ms |
| 32 | Tradition [6] | 19.29 ms |
| | Gaussian | 3.04 ms |

**Comparison of Entropy Latency.** To improve the efficiency of our **EA**, we did not use the traditional method to calculate the differential entropy, but instead calculated it conditioned on a Gaussian distribution for inference acceleration. We randomly generate a set of tensors to verify the efficiency of different manners to compute the differential entropy. To mitigate the impact of other factors, we repeated the calculation 100 times and collected the average results. As shown in Table 4, with the increasing batch size, traditional methods for computing differential entropy consume much more time, whereas calculating the differential entropy under a Gaussian distribution is nearly instantaneous. Basically, the computation of differential entropy presented in this work, which is conditioned on a Gaussian distribution, is almost ~60× to 70× faster than the traditional manner.

### 4.3 Comparison with Advanced Models

**Efficient SISR.** In this case, we compared our **EARFA** with EDSR [9], CARN [5], IMDN [40], LatticeNet [41], ESRT [42], SwinIR [13] and SRFormer [14] on five benchmark datasets. The quantitative results for three SR scales (SR×2, SR×3, and SR×4) are presented in Table 2, where we can observe that our **EARFA** shows the best compromise between SR performance and inference latency. Our **EARFA** has a more significant advantage in inference efficiency compared to Transformer-based models. For instance, our **EARFA** exhibits a latency of **35.40**ms with SR×4, while SRFormer [14] has a latency of **112.15**ms, which is 3.17 times of our **EARFA**. On the other hand, **EARFA** achieves PSNR gains of **0.19 dB** and **0.35 dB** on Urban100 [49] and Manga109 [4] respectively, compared to SRFormer [14]. These observations indicate that our **EARFA** obtains better SR performance with higher inference efficiency than other advanced SISR methods, striking a better balance between performance and efficiency.

**Super Lightweight SISR.** For this situation, we compare our **EARFA-light** with SRCNN [7], VDSR [8], DRRN [17], IDN [18], PAN [32], ShuffleMixer [43] and SAFMN [10] etc. As shown in Table 3, our **EARFA-light** also demonstrates superior performance for all scales and datasets. Specifically, **EARFA-light** maintains **209K** model parameters, which is lower than the best-performing super lightweight SR method known to us, i.e., SAFMN [10]. In terms

of SR performance, our **EARFA-light** presents better result than SAFMN [10] on the Urban100 [49] and Manga109 [4], by a margin of **0.25 dB** and **0.32 dB**, respectively. In this case, our **EARFA-light** provides a better tradeoff between SR performance and parameter scale. **EARFA-light** excels due to its high efficiency and superior performance, allowing it to deliver excellent SR results while being extremely lightweight.

### 4.4 Visual Results

The visual comparison between our models and other compared methods on Urban100 [38] with SR×4 is shown in Fig. 7. In some challenging scenarios, the previous methods may suffer blurring artifacts, distortions, or inaccurate texture restoration. In contrary, the proposed **EARFA** and **EARFA-light** can effectively mitigate these artifacts and preserve more structures and finer details. For example, in testing image "img_004", the reconstructed images of the previous methods are mostly have some problems such as blur, distortion, and poor detail recovery. And our proposed **EARFA** can restore the correct structure and details. We also observed this phenomenon in "img_012" and "img_092". This is primary because **EA** and **SLKA** provide our models with more informative inference and enhanced effective receptive fields.

## 5 CONCLUSIONS

In this work, we propose two novel components, i.e., **EA** and **SLKA** for ESISR tasks, and build an efficient SISR model **EARFA** based on these ingredients. The **EA** introduces differential entropy [6] into channel attention to relieve the issue that traditional pooling is inefficient in measuring the information of intermediate features. To improve the inference efficiency, we also propose an improved method for calculating differential entropy, which is constrained by Gaussian distributions. Besides, we also present an enhanced **SLKA** of LKA [28] with the aid of simple channel shifting [29], which can further expand the effective receptive field of the model, so as to endow the model with the wide-range perception capacity. Since channel shifting does not introduce additional parameters and the additional computational overhead (data movement) is negligible, the balance between SR performance and model efficiency can be significantly promoted. Extensive experiments have shown that our **EARFA** can achieve better SR results with higher efficiency, even compared to more advanced Transformer-based SISR models.

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
