# OpenReview forum: "Efficient Single Image Super-Resolution with Entropy Attention and Receptive Field Augmentation"
_acmmm.org/ACMMM/2024/Conference — MM2024 Poster_

### Official Review · Reviewer_XPZF · 2024-05-23

**Rating:** 5
**Confidence:** 3

**Summary:**

This paper proposes a new ESISR method, EARFA, in a Transformer-based framework. The differential entropy is used to define the significance of channel-wise features, a new attention mechanism is proposed, and the SLKA module is united. EARFA finally works on the super-resolution task with better performance while keeping lower model parameters.

**Strengths:**

Considering the computational complexity of MSA, a novel attention mechanism based on differential entropy is proposed, which can effectively solve the computational time-consuming problem of the Transformer architecture itself.I think Entropy Attention (EA) is a good idea to use the prior knowledge of the Gaussian distribution to effectively evaluate the importance of features.  EA is able to have a similar functionality to the pooling operation by simply obtaining the weights of each channel feature while reducing the loss of information in the transmission process.
In addition, the time-consuming processing of continuous data by traditional methods is solved by aligning features with Gaussian distributions as much as possible. To some extent, the problem of higher complexity of Transformer-based models is solved.
To further obtain LARGE receptive fields, the LKA module is improved by replacing the original point-wise convolution with a shifting convolution.
Better results compared to some SOTA methods;
Finally, from the experimental results, comparing with some SOTA methods, the latency and results of EARFA are superior on the same equipment.

**Limitations:**

The motivation could be written more clearly in the INTRODUCTION;
With an effectively enlarged receptive field, how to ensure that local information can also be effectively extracted;

**Suitability:**

2

---

### Official Review · Reviewer_Ro5E · 2024-05-24

**Rating:** 2
**Confidence:** 3

**Summary:**

The paper presents an approach to single image super-resolution (SISR) by introducing Entropy Attention (EA) and Shifting Large Kernel Attention (SLKA) mechanisms within theproposed EARFA model. The idea of integrating information theory into attention mechanisms iseffective in image super-resolution.

**Strengths:**

1. The theoretical approach is justified through the detailed explarnation of EA in the methodology section
where the authors discuss the use of differential entropy. This is quite innovative
2. By combining EA and SLKA blocks, the model has achieved remarkable performance in super-resolution
tasks and has lower time consumption than new SR modelssuch as SRFormer.

**Limitations:**

1. In convolutional neural networks, combining spatial attention and channel attention is not innovative, and
it has been widely used since 2018 (just like: CBAM: Convollutional Block Attention Module, and Dual Attention Network). However, the paper did not mention or conductt an ablation study on these works
2. Ablation study about the attention mechanism is not sufficient. The ablation study of channel attention
should include at least some of the most classic methods (SE, ECA,SK, CBAM). and provide more detailed comparison tables (performance and time consumption comparison).
3. For the name of Shift Large Kernel Attention (SLKA) moddule, 5x5 depth wise conv doesn't seem to be
"Large Kernel". Shift-conv could make small convolutional kernel work like with large kernel effects. The purpose of shift-conv is to replace the large kernel.

**Suitability:**

3

---

### Official Review · Reviewer_ydEs · 2024-05-25

**Rating:** 4
**Confidence:** 2

**Summary:**

The paper presents an efficient single image super-resolution (SISR) model called Entropy Attention and Receptive Field Augmentation (EARFA). The Entropy Attention(EA) introduces differential entropy into channel attention to relieve the issue that traditional pooling is inefficient in measuring the information of intermediate features. To improve the inference efficiency, the authors also propose an improved method for calculating differential entropy, which is constrained by Gaussian distributions. Besides, they also introduce the channel shifting strategy (proposed by Zhang et al.) into the LKA (proposed by Guo et al.) to get the SLKA to expand the effective receptive field. The proposed method achieves better results compared to several state-of-the-art methods.

**Strengths:**

1. The paper introduces a new approach to address the tradeoff between model efficiency and SR performance in SISR tasks. The combination of entropy attention and shifting large kernel attention is a novel contribution.
2. In general, the paper provides a clear description of the proposed method, including the overall structure of EARFA and the components of EA and SLKA.

**Limitations:**

The experiments presented in the paper are not sufficiently comprehensive, as they lack comparisons to some recently published, high-quality methods, such as Omni-SR, which was introduced at CVPR 2023. Based on the Omni-SR paper, it appears that the computational cost, quantitative results, and some of the visual effects for the same image presented in the EARFA paper are outperformed by Omni-SR. Consequently, the claimed superiority of the proposed EARFA model over state-of-the-art approaches is questionable without further comparisons and evaluations against the latest techniques in the field. To solidify the contributions and merits of EARFA, it is essential to conduct a more thorough experimental analysis that includes a wider range of cutting-edge methods, ensuring a fair and comprehensive assessment of the proposed approach.

**Suitability:**

2

---

### Official Review · Reviewer_bE2d · 2024-06-06

**Rating:** 3
**Confidence:** 3

**Summary:**

This paper proposes a novel model for efficient single image super-resolution (SISR) called EARFA, which consists of an entropy attention (EA) mechanism and a shifting large kernel attention (SLKA) module. The EA aims to improve the information entropy of intermediate features conditioned on a Gaussian distribution to provide more informative inputs. The SLKA extends the receptive field using channel shifting while avoiding heavy computations. The authors claim their model achieves faster inference than transformer-based methods while maintaining competitive SR performance.

**Strengths:**

The paper is well-written and clearly describes the proposed method.

Proper comparisons are made against state-of-the-art efficient and transformer SISR methods.

**Limitations:**

Motivation and Problem Statement: The motivation behind EARFA is not clearly articulated. The introduction suggests that the core challenge is to mimic the non-local perception and expanded receptive field of transformer models while reducing computational overhead. However, this reasoning is questionable, as many recent transformer models for (efficient) SISR employ window-based self-attention with limited receptive fields. The authors do not sufficiently justify how EARFA fundamentally addresses this challenge better than existing approaches.

Limited Novelty: The proposed components seem to be combinations of existing techniques rather than novel ideas. SLKA is similar to the shifting operations explored in works like ELSN. While EA's motivation of using entropy for attention is unclear, it appears to be a straightforward adaptation of ideas from other domains without a clear rationale for improving efficiency in SISR tasks.

Empirical Analysis: The experimental results do not convincingly demonstrate superior efficiency compared to existing efficient SISR models. For instance, while EARFA achieves limited performance gains over methods like ShuffleMixer, its inference latency is nearly twice as high. From the perspective of pursuing an efficient SISR solution, the paper's rationale and methodology appear inconsistent and poorly justified.

Relevance to Multimedia: The research topic of efficient single image super-resolution seems tangentially related to the core themes of the ACM Multimedia conference. The paper primarily focuses on a unimodal computer vision task without explicit connections to multimedia or multimodal processing.

**Suitability:**

1

---

### Meta-Review · Area_Chair_v4qK · 2024-06-28

**Recommendation:** Accept (Poster)
**Confidence:** 4

**Metareview:**

This paper introduces a new approach to address the trade-off between model efficiency and model performance in the task of single image super resolution. The paper is well-written, and its technical contribution is sound. The rebuttal effectively addressed the major concerns raised by all reviewers. After the rebuttal, all reviewers reached a consensus to borderline accept the paper. The AC has carefully reviewed the paper, its rebuttal, and the discussion, and believe the paper has merit and recommend its acceptance. The AC urges the authors to revise their paper by taking into account all the reviewers' feedback, such as improving the writing about the motivation and core contributions and including the additional results provided in the rebuttal, to further strengthen their paper.